# P2X4 receptors mediate induction of antioxidants, fibrogenic cytokines and ECM transcripts; in presence of replicating HCV in *in vitro* setting: An insight into role of P2X4 in fibrosis

Sobia Manzoor[1,2]*, Madiha Khalid[2], Muhammad Idrees[1]

1 Center of Excellence in Molecular Biology (CEMB), University of Punjab, Lahore, Pakistan, 2 Atta-ur-Rahman School of Applied Biosciences (ASAB), National University of Science and Technology (NUST), Islamabad, Pakistan

* dr.sobiamanzoor@asab.nust.edu.pk, lcianunique@yahoo.com

## Abstract

### Background & aims

Major HCV infections lead to chronic hepatitis, which results in progressive liver disease including fibrosis, cirrhosis and eventually hepatocellular carcinoma (HCC). P2X4 and P2X7 are most widely distributed receptors on hepatocytes.

### Methods

Full length P2X4 (1.7kb) (*Rattus norvegicus*) was sub cloned in mammalian expression vector pcDNA3.1+. Two stable cell lines 293T/P2X4 (experimental) and 293T/ NV or null vector (control) were established. Both cell lines were inoculated with high viral titers human HCV sera and control human sera. Successfully infected cells harvested on day 5 and day 9 of post infection were used for further studies.

### Results

The results revealed a significant increase in gene expression of P2X4 on day 5 and day 9 Post -infection in cells infected with HCV sera compared with cells inoculated with control sera. Quantitative real time PCR analysis revealed that HO-1 was significantly upregulated in presence of P2X4 in HCV infected cells (P2X4/HCV) when compared with control NV/ HCV cells. A significant decrease was observed in expression of Cu/ZnSOD in presence of P2X4 in HCV infected cells compared to control NV/HCV cells. However, expression of both antioxidants was observed unaltered in cells harvested on day 9 post infection. Gene expression of angiotensin II significantly increased in HCV infected cells in presence of P2X4 on day 5 and day 9 of post infection when compared with control NV/HCV cells. A significant increase in gene expression of TNF-α and TGF-β was observed in HCV infected cells in presence of P2X4 on day 9 post infection in comparison with control (NV/HCV cells).

**Data Availability Statement:** All relevant data are within the manuscript and its Supporting Information files.

**Funding:** AM and MIK received funding from the Higher Education Commission of Pakistan. The funders had no role in study design, data collection and analysis, decision to publish, or preparation of the manuscript.

**Competing interests:** The authors have declared that no competing interests exist.

However, gene expression of adipokine leptin was not affected in both experimental (P2X4/HCV) and control (NV/HCV) groups on day 5 and day 9 of post infection. Extracellular matrix proteins, laminin and elastin genes expression also significantly increased in presence of P2X4 (HCV/P2X4) on day 9 of post-infection compared to control group NV/HCV cells.

## Conclusion

In conclusion, these findings constitute the evidence that P2X4 receptors in the presence of HCV play a significant role in the regulation of key antioxidant enzymes (HO-1, Cu/ZnSOD), in the induction of proinflammatory. cytokine (TNF-α), profibrotic cytokine (TGF-β) vasoactive cytokine (angiotensin II). P2X4 also increases the expression of extracellular matrix proteins (laminin and elastin) in the presence of HCV.

## Introduction

Hepatitis C virus (HCV) is an enveloped virus with a single positive-sense strand RNA genome, isolated in 1989, belongs to the family Flaviviridae [1–8]. There are two types of infections namely; acute and chronic, acute HCV infection found to be persistent in about 85% of cases [9]. All over the world, 3.3% of the population is infected while 10% of the Pakistani population is chronically infected with HCV [10–13].

HCV has a single-stranded RNA genome comprising of 96000bp long, the polyprotein is posttranslationally cleaved by both viral/cellular proteases to produce about 10 polypeptides [13–18]. Major HCV infections lead to chronic hepatitis, which results in progressive liver disease including fibrosis, cirrhosis, and eventually hepatocellular carcinoma (HCC) [4, 12]. Approximately 30% of chronically infected patients develop liver fibrosis and cirrhosis. It is estimated that there is an annual risk of 1% to 4% among patients with cirrhosis to develop hepatocellular carcinoma (HCC) [5–7]. There is a robust link reported between chronically infected patients with HCV having genotype 3a and HCC (63.44% of tested HCC patients) in the Pakistani population [10].

The disease is manifested by viral replication within hepatocytes, generation of reactive oxidative species (ROS), triggering of innate and specific immune responses, necroinflammation with the destruction of hepatocytes, and slowly progressive development of fibrosis. Several studies are being focused to find out the association between viral-induced signaling pathways that lead towards chronic disease manifestation [13, 19]. Though the characteristics and complications of HCV are well identified, the molecular mechanisms of HCV-induced pathogenesis are yet to be fully understood [2, 19]. However, oxidative stress has emerged as a key player in the development and progression of HCV-induced pathogenesis including inflammation, fibrosis, altered gene expression. and hepatocellular carcinoma (HCC) [8]. In Purinergic signaling, adenosine 5′-triphosphate (ATP) and adenosine act as extracellular signaling molecules. There are seven subtypes of P2X ion channel receptors (P2X1-7). Adenosine 5′-triphosphate (ATP) and its metabolites constitute important signaling pathways that regulate a broad variety of biological processes [20, 21]. In the liver, numerous pieces of evidence support a significant role for extracellular ATP-induced signaling pathways in the control of tissue homeostasis. Like, ATP and other nucleotides are physiologically detected in extracellular fluids within the liver in various mammalian species including humans and rodents. Intercellular purinergic signaling within hepatocytes is important for the regulation of key cellular processes including cell survival and death [22].

It is observed that P2X4 and P2X7 are the most widely distributed receptors on liver cells. Several P2 receptors are being studied on liver hepatocytes and other functionally specialized cells and it is also noteworthy to know the effect of ATP-mediated changes on liver physiology by exploiting P2XR. [23, 24]. There is a number of agonists and antagonists that have been studied to characterize the presence of these receptors on liver hepatocytes by using different *in vitro* and *in vivo* models.

The role of purinergic signaling in HCV-induced liver fibrosis is not yet known. As these are the active receptors on hepatocytes and purinergic receptors could be active targets for drug implication. To investigate the role of purinergic mechanisms, we investigated the role of P2X4 receptor involvement in HCV-induced liver fibrosis. The vector expressing purinergic receptor P2X4 and HCV sera were introduced externally in H239T to examine their effect on cellular genes specific for liver fibrosis. This study makes a new way for understanding how HCV exploits cellular receptors to interrupt normal homeostasis.

## Material and methods

### Cloning of P2X4 and development of cell culture-based system using vector carrying P2X4 and human HCV serum

**Subcloning of P2X4 in a mammalian expression vector pcDNA3.1+ (Construction of Expression plasmid).** Full-length P2X4 (1.7kb) cloned in plasmid PCR 3.1(pCR3.1/P2X4) was kindly provided by Dr. Ishtiaq Qadri King Abdulaziz University, Jeddah, Saudi Arabia P2X4 was excised from vector PCR 3.1 and cloned in vector pcDNA3.1+ to obtain expression plasmid, P2X4/ pcDNA3.1+.

Establishment of stable cell lines:

Two different cell lines 293T/P2X4 (stably expressing P2X4 protein, Experimental), and 293T/pcDNA3.1+ or 293T/NV (vector alone without any insert/ null vector) were established to study the role of P2X4 in the presence of HCV replicating RNA.

The followings steps were involved:

**Linearization and purification of digested of plasmids.** Plasmids, P2X4/ pcDNA3.1+. and pcDNA3.1+ were linearized before transfection to facilitate their stable integration in 293T cells genome. Restriction enzyme *Bgl II* (Fermentas) was employed to linearize pcDNA3.1+/P2X4 and pcDNA3.1+. Before using *Bgl II*, it had been confirmed using software NEBcutter V2.0 that this enzyme was quite safe for important sequences required for protein synthesis of the inserted gene. *Bgl II* cut the pcDNA3.1+ only at single site (nucleotide number 13, far away from CMV promoter. *Bgl II* did not cut at any nucleotide site of inserted gene P2X4 (using software NEBcutter V2.0). Plasmids were also quantified using NanoDrop® (Spectrophotometer) (ND-1000) to facilitate the optimization of linearization reaction.

Digested plasmids were run on 1% agarose gel, 80V for 90 min and analyzed under UV-light. The gel slices containing linearized/ digested products (psP2X4 and pcDNA 3.1+) were cut with sterilized blade. Plasmids were purified according to the protocol of Fermentas DNA Extraction Kit (Cat # K0513). 5µl of purified plasmid were run on 1% agarose gel with λHin-dIII digested marker to check concentration for transfection. Concentration was also checked on nanodrop.

**Cell line and culture condition.** The cell line used in this study, HEK 293T (HEK, human embryonic kidney cell line) was kindly provided by Dr. Ishtiaq Qadri King Abdulaziz University, Jeddah, Saudi Arabia (). 293T cell line was cultured in Dulbecco Modified Eagle's Medium (DMEM) supplemented with 100U/ml of penicillin and 100ug/ml of streptomycin and 10% heat inactivated fetal bovine serum. After transfection with pcDNA3.1+ with a zeocin resistance cassette (Invitrogen) that contained full length P2X4 or vector control (293T/P2X4 and

293T/Null vector), 293T cell lines were maintained in medium supplemented with 1mg/ml G418 for more than one month for the selection of stable clones. 293T stable cell lines were grown in DMEM supplemented with 500ug/ml G418. Cells were maintained at 37˚C in a humidified environment containing 5% $CO_2$ in a cell culture incubator. The cells were sub-cultured on every 3 days.

## DNA transfections

293T cells were prepared for transfection by plating onto 4 Petri-dishes (60mm) at the time of subculture, 2days before transfection in DMEM containing appropriate supplements. They were transfected with plasmid pcDNAP2X4 (experimental) and pcDNA3.1 vector alone (control).

After 2days cells became 70–80% confluent and transfections were carried out using Lipo-fectamineTM 2000 (Invitrogen, Cat No.11668-019) following the manufacturer's instructions.

Two different concentrations of purified, linearized plasmid DNA (2µg DNA+6µl Lipofec-tamine and 4µgDNA+10µl Lipofectamine) were used for optimization of transfection reaction. At 48 hrs post transfection, cells were sub cultured into 25cm2 culturing flasks and 60mm petri-dishes and were grown in DMEM supplemented with selective agent G418 (1mg/ml).

**Selection of stable cell lines.** After successful transfections, cell lines 293T/P2X4 and 293T/Null vector (NV) were continuously grown in complete DMEM supplemented with G418 (1mg/ml). Cells were subcultured on every day 4–5 depending upon confluency of cells. At the same time 293T cells without any transfection were also grown in the presence of selective reagent G418 (1mg/ml) as a control for optimization of selection of resistant stable clones. The cells in this flask (control) were also subcultured, old media was replaced with fresh one supplemented with G418 (1mg/ml).

293T cells in this control culture (without any transfection but continuously receiving selective agent G418 at the same concentration of transfected cell lines) were died after 20-25days. Transfected cell lines were grown in G418 (1mg/ml) continuously for more than 1 month (35days). After that, cells were sub-passage, approximately 25% grown for 2–3 day in the presence of G418 (1mg/ml). Single colonies were isolated using sterile filtered tips employed on sterile pippett, under microscope and sterile conditions. Single clones were transferred to 24-well culturing plates, grown in the presence of G418 (500µg/ml). After 15 days single clones were trypsinized and shifted into 6- well plates. Upon 70% confluence, single clones with stable expression were sub-passage into 25cm$^2$ culturing flasks. Expression of P2X4 was verified by reverse transcriptase PCR (mRNA) and western blotting (protein) and highest expressing clones were selected for further experiments.

**Verification of stable cell lines.** *Reverse Transcription–Polymerase Chain Reaction (RT-PCR) analysis*. Previously reported primers were used for detection of transfected P2X4 mRNA (Doctor et al., 2004) while for detection of vector primers were designed at sequence T7 promoter and BGH reverse priming site using Primer3 software (http://bioinformatics.weizmann.ac.il/cgi-bin/primer/primer3.cgi). **Table 1**.

**Table 1. List of Primer sequences for cloning.**

| Primer Name | Primer Sequence: 5'-3' Sequence |
| --- | --- |
| PX4T- F | CGTGGCGGACTATGTGATT |
| PX4T-R | GTGATGTTGGGGAGGATGTTC |
| T7 | GTAATACGACTCACTATAGGG |
| BGH | TAGAAGGCACAGTCGAGG |

**PCR amplification of transfected P2X4.** PCR conditions were 94˚C (2 minutes), 35 cycles [94˚C (30 seconds), 58˚C (30 seconds) and 72˚C (1 minute)], and 72˚C (10 minutes).

The amplified product was carefully excised from the gel, using a sterilized surgical blade and transferred into a sterile eppendorf and eluted from agarose gel using gel extraction kits (Fermentas; Cat No.K0513) by following the manufacturer's instructions.

The identity of amplified product was confirmed by sequencing using an automated sequencer.

Sequences were analyzed manually by using Chromas software version (v 1.45). Homology studies of the nucleotide sequences of amplified and sequenced P2X4 with known nucleotide sequence was done through standard Basic Local Alignment Search Tool software available at NCBI website. The sequence of P2X4 showed 100% homology with reference sequence.

**PCR amplification of transfected null vector.** PCR conditions were 95˚C (3minutes), 30 cycles [95˚C (30 seconds), 52˚C (30 seconds) and 72˚C (45 seconds)], and 72˚C (10 minutes). Amplified product of expected size was obtained.

*Western blot analysis*. To study protein expression of P2X4 in stable cell line, 100μg of total protein were loaded in each well on 10% SDS–PAGE gels and electrophoretically blotted onto a nitrocellulose membrane (Bio-Rad).

The expression level of P2X4 was determined using antibody specific to P2X4. The protein concentration of cell extracts was determined using a protein assay reagent (Bio-Rad). The membranes were blocked for over–night at 4˚C with phosphate–buffered saline containing 5% skim milk. After being washed with 1xPBS containing 0.1% Tween 20, the membranes were incubated with primary antibody specific to P2X4 (Chemicon International, Cat no.AB5226) for over–night at 4˚C. Wash the cells three times with 1X PBST and treated with secondary antibody (Chemicon International, Cat no.AP132A) for over–night at 4˚C. After being washed with PBST proteins expressions were evaluated using 5-bromo-4-chloro-3-indolyl-phosphate/ 4-nitro blue tetrazolium chloride / (BCIP/ NBT) solution.

Western blot and RT-PCR analysis were performed at 48h and 96 h of post transfection and after that continuous selection with G418 for of 1 month. Singles clones were also analyzed at RT-PCR level and through Western blot analysis. Highest null vector (NV) mRNA express-ing and P2X4, mRNA and protein expressing clones (transfected with 4μg plasmid DNA) were selected for further studies.

*Viral inoculation and sample collection*. Stable cell lines 293T/P2X4 (stable clone transfected with and stably over expressing P2X4 protein) and 293T/NV (stably expressing vector alone or null vector) were grown in 60mm Petri-dishes to 70% confluency and 40% confluency in two different groups.

Required confluency obtained after 2days of sub-culturing, cells were washed with 1XPBS and 1ml of complete DMEM supplemented with G418 (500μg/ml) was added in each dish.

Cells were inoculated with human HCV sera and control sera (240μl serum was mixed gently in 1ml complete DMEM medium/ per dish). Control sera and sera from chronically infected HCV patients (local population) with high viral titers and genotype 3a were obtained from Diagnostic Research Laboratories of centre (CEMB). Control sera (normal sera) were obtained from subjects negative for Hepatitis A, B and C, CMV and HIV.

The viral load in the used HCV sera was quantified by real time PCR using HCV Real-TM Quant kit (REF, TVI-96/2FRT C SP, Sacace Biotechnologies, Italy) following the manufactur-er's instructions. After 40-48h of inoculation, adherent cells were washed two times with IX PBS to get rid of the remaining infection serum. Incubation was continued in complete DMEM containing G418 (500μg/ml). The cells were maintained overnight at 37˚C in a humid-ified environment containing 5% $CO_2$. On day 5th RNA was isolated from cells using Gentra

RNA Isolation Kit (Puregene, Minneapolis, MN 55441, USA) according to the provided protocol.

Formula for the calculation of HCV RNA concentration

Following formula was used to calculate the concentration of HCV RNA of each sample.

Cy3 STD/Res X Coefficient IC = IU HCV/ml. Fam Std/Res

Co-efficient, IC = internal control, which is specific for each lot.

Fluorescence is observed in Real Time on the Cy3 channel for HCV RNA and FAM channel for Internal Control.

Successfully inoculated RNA samples (with significant HVC titers) were quantified and further processed for cDNA synthesis.

On day 9th, RNA was isolated, viral titers were quantified through real time PCR. RNA samples with significant viral titers were further processed for cDNA synthesis. Inoculated cell lines showed significant HCV viral titers on day 5 and day 9 but not on day 14. So, cDNA of four different cell lines P2X4/HCV, P2X4/NR, NV/HCV and NV/NR after day 5 of inoculation and day 9 of inoculation were processed for further study.

*Study the role of P2X4 receptor (Purinergic Signaling) in HCV induced liver fibrosis*. Fibrosis is multifactorial involving interaction of various cellular markers. Therefore, in order to study the role of P2X4 receptor (Purinergic Signaling) in HCV induced liver fibrosis, we selected eight potent genes from different categories including oxidative stress (Cu/Zn-SOD, HO-1) cytokines (TNF-α, TGF-β1, Leptin, Angiotensin II) and finally extracellular matrix proteins (Laminin, Elastin) and above mentioned cell culture based system successfully inoculated with human HCV sera (i.e cell lines P2X4/HCV, P2X4/NR, NV/HCV and NV/NR) by following different steps.

*Designing of PCR primers for selected markers*. Sense and anti-sense primers for the amplification of different markers including Cu/Zn-SOD, HO-1, TNF-α, TGF- β1, Leptin, Angiotensin II Laminin and Elastin were designed using primer3 software.

Following gene-specific primers were used for PCR amplification from cDNA. **Table 2**.

**Table 2. List of primers for the selected genes.**

| S. No. | Primer ID | Primer Sequence: 5'-3' Sequence |
|---|---|---|
| 1 | Angiotensin II-F | CACGCTCTCTGGACTTCACA |
| 2 | Angiotensin II-R | GCTGTTGTCCACCCAGAACT |
| 3 | TGF- β1-F | TATCGACATGGAGCTGGTGA |
| 4 | TGF-β1-R | TGGGTTTCCACCATTAGCAC |
| 5 | HO-1-F | AGGTCATCCCCTACACACCA |
| 6 | HO-1-R | GTTGGGGAAGGTGAAGAAGG |
| 7 | Leptin-F | ACGTGCTGGCCTTCTCTAAG |
| 8 | Leptin-R | ACCTGGAAGCCAGAGTTCCT |
| 9 | Cu/Zn SOD-F | GGGGAAGCATTAAAGGACTG |
| 10 | Cu/Zn SOD-R | AATAGACACATCGGCCACAC |
| 11 | TNF -α-F | TCCTTCAGACACCCTCAACC |
| 12 | TNF -α-R | CAGGGATCAAAGCTGTAGGC |
| 13 | GAPDH-F | ACCACAGTCCATGCCATCAC |
| 14 | GAPDH-R | TCCACCACCCTGTTGCTGTA |
| 15 | ELAS–F | CAGGAGTTGGTGGCTTAGGA |
| 16 | ELAS- R | CTGGAGCCTTGGGCTTAACT |
| 17 | LAM- F | CACCAAGTCCTGTCACCTGT |
| 18 | LAM- R | CTGGTGTGGAACTTGAGACG |

The samples were preheated at 94˚C for 45seconds and then run 35 cycles with the following parameters: at 94˚C for 1 minute, 57˚C (Cu/Zn-SOD, HO-1,Laminin and Elastin), 55˚C (TNF-**α**), 59˚C (TGF-β1), for 45 seconds and 72˚C for 1 minute. Final extension was done for 10 minutes at 72˚C. The amplified PCR products of expected size were obtained.

*Transcriptional expression analysis.* The gene expression of Cu/Zn-SOD, HO-1, TNF-α, TGF-β, Leptin, Angiotensin II, Elastin, Laminin and P2X4 (Transfected) was analyzed in cell lines 293T/P2X4-HCV, 293T/P2X4-NR, 293T/NV-HCV and 293T/NV- NR (after day 5 and day 9 post inoculation) using real time Real Time PCR.

Real time PCR reactions were run on Cepheid smart cycler II (France) at same PCR Profiles which were used to optimized PCR products at conventional PCR (ABI-2700) thermal Cycler).

*Statistical analysis.* All the data was analyzed by using GraphPad Prism 5 software Version 5.02 and data was represented as mean ± SEM. ANOVA test was used to compare the parameters. A significant data was denoted with a *p*-value of less than or equal to 0.05.

## Results

### Sub cloning of P2X4 gene in expression vector

To examine the effect of P2X4 on genes reported to induce liver fibrosis, in presence of all HCV structural and nonstructural proteins, it was subcloned in mammalian expression vector pcDNA3.1+. Full length P2X4 (1.7kb) (*Rattus norvegicus*) cloned in plasmid pCR 3.1(pCR3.1/ P2X4) along with its sequenced chromatogram was kindly provided by Dr. Ishtiaq Qadri King Abdulaziz University, Jeddah, Saudi Arabia. P2X4 was excised from vector pCR 3.1 and cloned in mammalian expression vector pcDNA3.1+ to obtained plasmid, P2X4/ pcDNA31+. **Figs 1–4.**

**Establishment of stable cell lines.** HEK 293T cell lines stably expressing P2X4 receptor (293T/P2X4) and vector alone or null vector (293T/NV) were established by transfection of vector carrying P2X4 gene and vector alone respectively. Highest null vector (NV) mRNA expressing and P2X4, mRNA and protein expressing clones were selected for further studies. **Figs 5–9.**

### Infection of stable cell lines with naturally occurring human HCV

To examine the role of P2X4 receptor in HCV induced liver fibrosis (P2X signaling); sera from chronically HCV- infected patients with high viral titers were used to infect stable cell lines

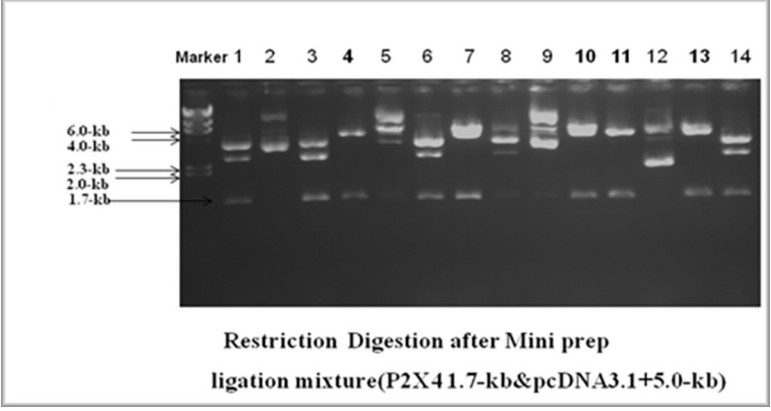

**Fig 1. Confirmation of P2X4 gene in vector pcDNA3.1+ Marker: λHindIII digested marker.** Lanes 4, 10, 11, and 13 showing digested plasmid with the release of insert.

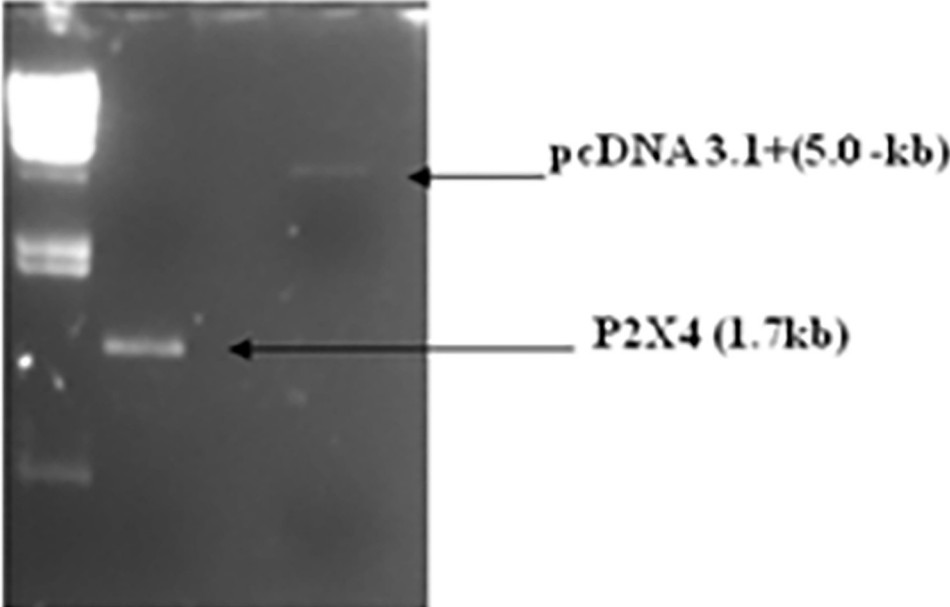

**Fig 2. Confirmation of P2X4 gene in vector pcDNA3.1+ pcDNA 3.1+ mammalian expression vector (5.0-kb), Insert P2X4 (1.7-kb).**

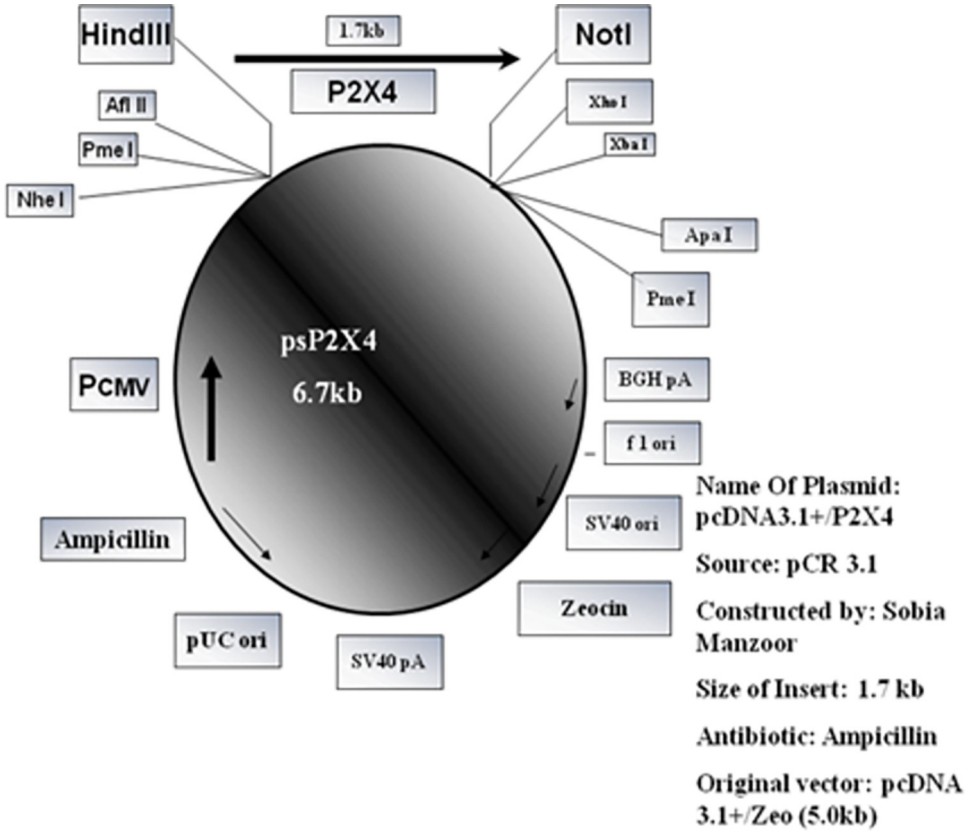

**Fig 3. Map of construct pcDNA3.1+/P2X4.**

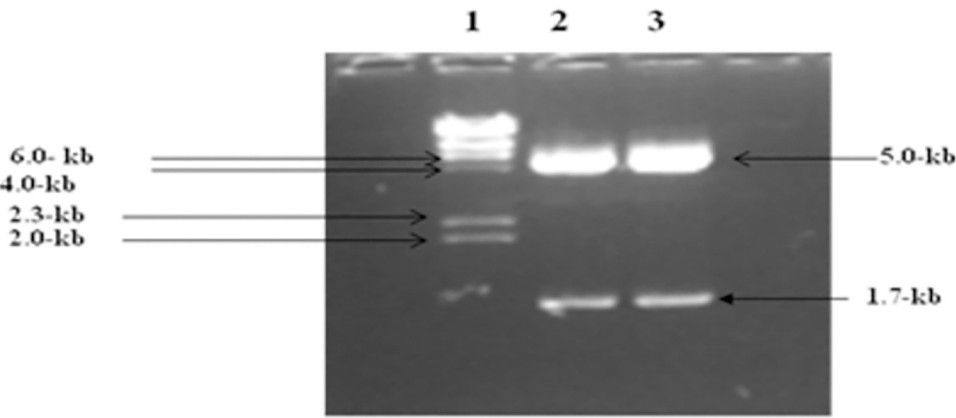

**Fig 4. Plasmid pcDNA3.1+/P2X4 restriction digestion after mini prep.** Lane 1 showed: λHindIII digested marker, Lanes 2 and 3 showing digested plasmid with release of insert, pcDNA 3.1+ mammalian expression vector (5.0-kb), Insert P2X4 (1.7-kb).

293T/P2X4 and 293T/NV. HCV-RNA was detected in infected cells using quantitative RT-PCR. Fig 10 depicts successfully infected clones at day 5 and day 9 of post infection were used for further studies. **Fig 10.**

## P2X4 gene expression in presence of HCV

HEK-293 stably expressing P2X4 was inoculated with patient serum having HCV; cells were incubated with virus to measure the effect of HCV on expression of P2X4 receptor. Expression of P2X4 gene was significantly increased at day 5 and day 9 of post infection in cells infected with HCV serum when compared with control cells (inoculated with normal human serum). Results were obtained from 3 individual experiments with replicates sample in each experiment are shown. **Fig 11.**

## P2X4 receptor signaling in HCV induced liver fibrosis

It is assumed that HCV have effect on purinergic signaling, the combined effect of HCV on P2X4 receptor expression and various cellular genes that are involved in liver fibrosis were examined. The expression of these genes were measured post infection with HCV serum in stably expressing HEK-293. **Fig 12.**

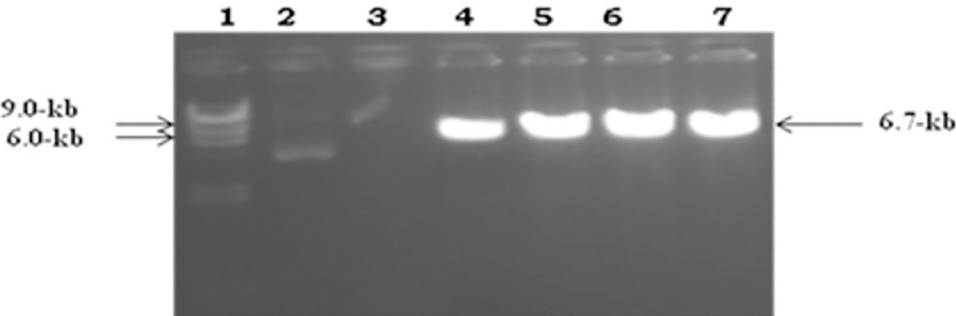

**Fig 5. Linearization of vector pcDNA3.1+/P2X4 before transfection.** Lane 1 showed: λHindIII digested marker, Lanes 4, 5, 6 and 7 showing digested linerized plasmid pcDNA3.1+/P2X4 (6.7-kb).

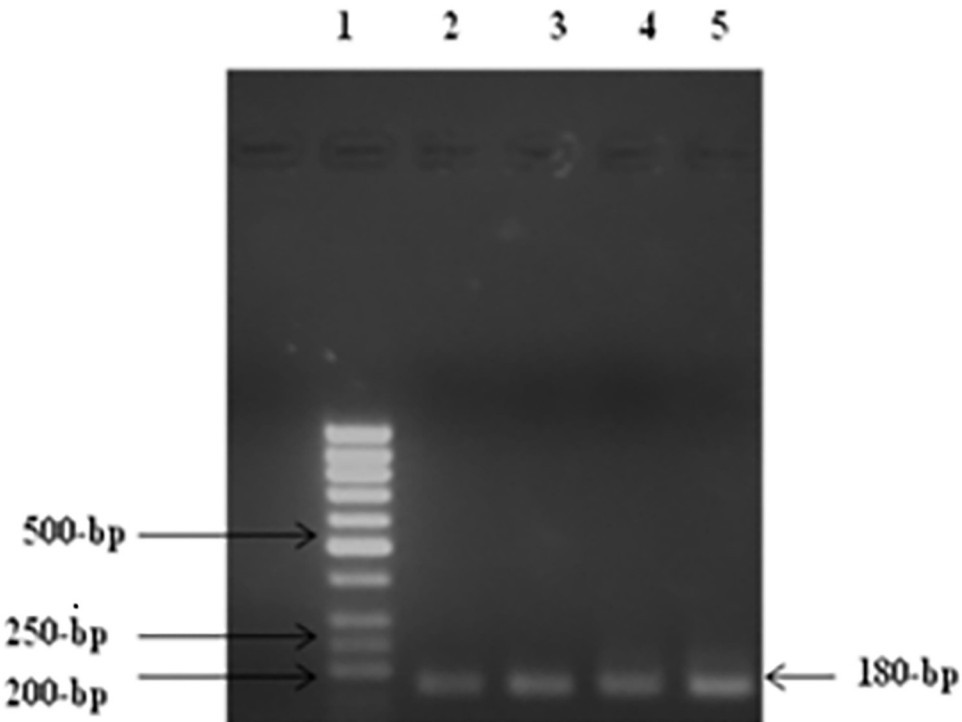

**Fig 6. RT-PCR of cell line 293T/P2X4.** Lane 1 showed: 50-bp DNA size marker, Lanes 2 &3 showing amplified product of transfected P2X4 (390-bp).

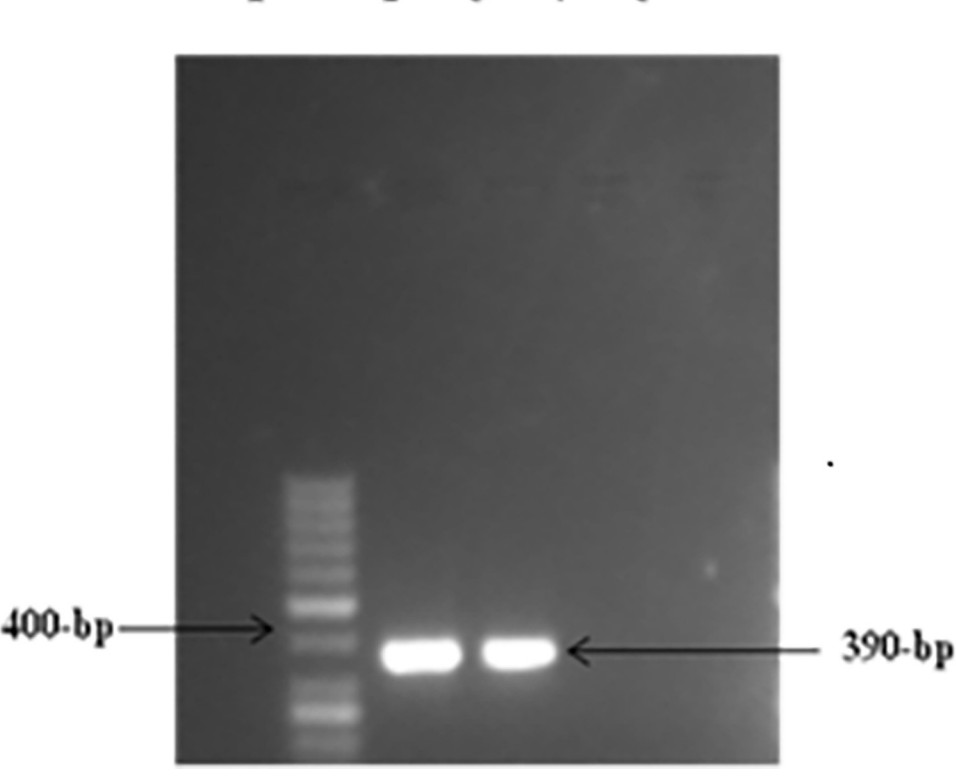

**Fig 7. PCR of cell line 293T/NV.** Lane 1 showed: 50-bp DNA size marker, Lanes 2,3.4 and 5 showing amplified product for NV (180-bp) (T7and BGH).

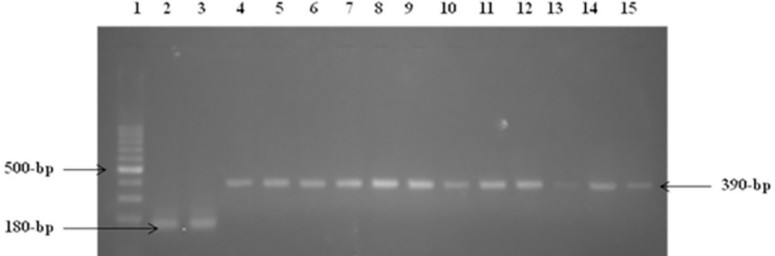

**Fig 8. RT-PCR of single clones of cell lines 293T/P2X4 and 293T/NV.** Lane 1 showed: 100-bp DNA size marker, Lanes 2 &3 showing amplified product of NV (180-bp), Lanes4-15 showed amplified product of transfected P2X4 (390-bp).

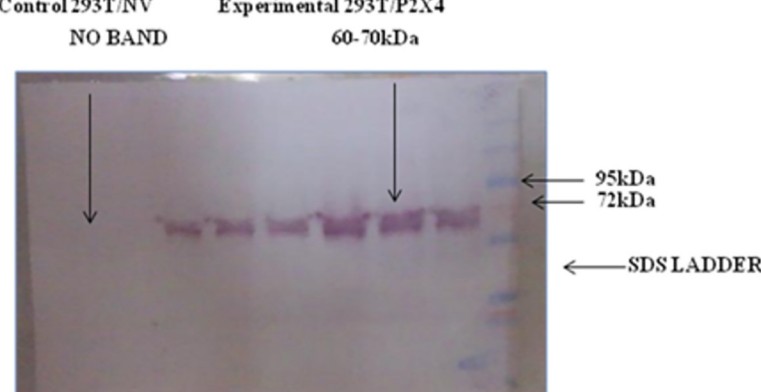

**Fig 9. Protein expression analysis of P2X4 receptor.** Western blot of P2X4 receptor (Protein lysate of selected single clones of cell lines 293T/P2X4 and 293T/NV). The predicted molecular mass of P2X4 is ~ 46kDa, but it is generally detected near 60–70 kDa because of glycosylation.

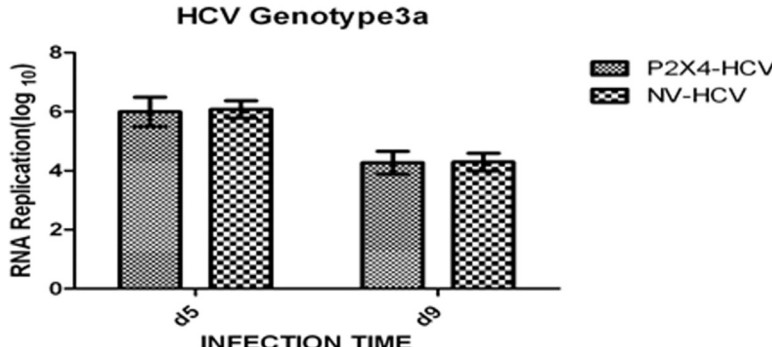

**Fig 10. HCV RNA was quantified by RT-QPCR after infection in stable cell lines.** All values are expressed as mean ± SEM *$P \leq 0.05$ vs. control NV/HCV.

## P2X4 in regulation of antioxidants: Mediate regulate

HCV disrupts the cellular balance of oxidant and antioxidants. Antioxidant effect was measured in stably expressing cell line in the presence of HCV. Antioxidant heme oxygenase-1 (HO-1) is significantly up regulates in presence of P2X4 in HCV infected cells when compared

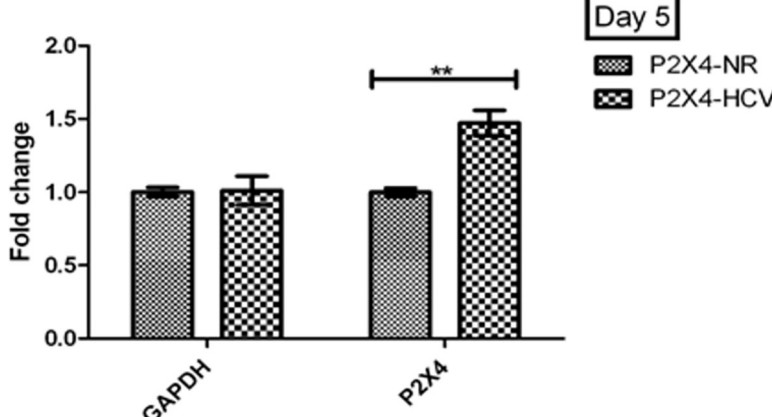

**Fig 11. Expression of P2X4 gene at day 5 of post infection with HCV.** All values are expressed as mean ± SEM, *P≤0.05 vs. control P2X4-NR cells.

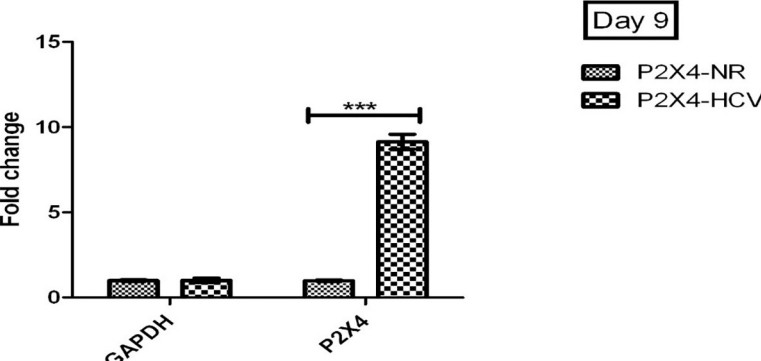

**Fig 12. Expression of P2X4 gene at day 9 of post infection.** All values are expressed as mean ± SEM, *P≤0.05 vs. control P2X4-NR cells.

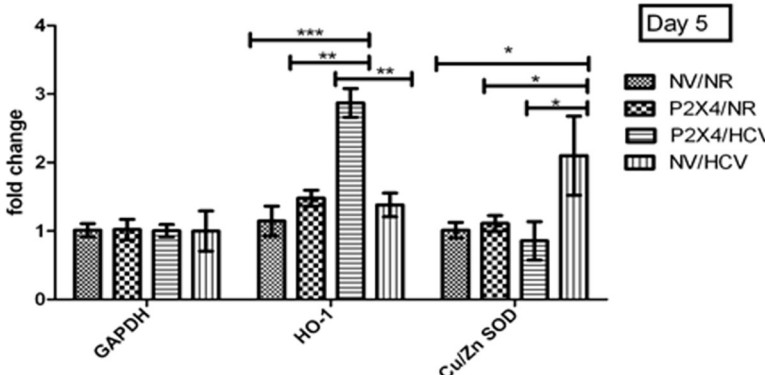

**Fig 13. Expression of antioxidants on day 5 of post infection.** All values are expressed as mean ± SEM, *P≤0.05 vs. control NV/HCV.

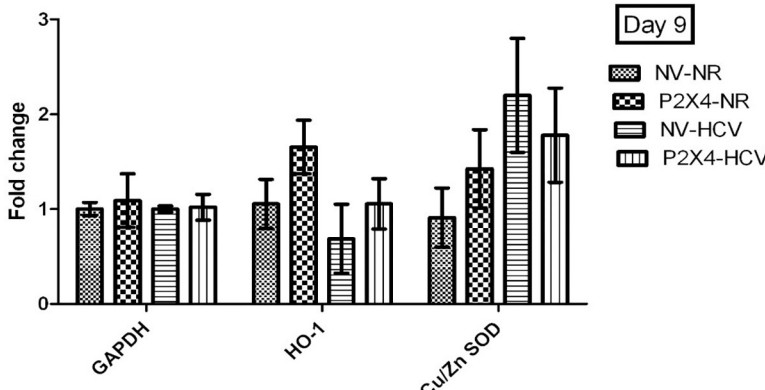

**Fig 14. Expression of antioxidants on day 9 of post infection.** All values are expressed as mean ± SEM, *P≤0.05 vs. control NV/HCV.

with control NV/HCV cells. A significant decrease was observed in expression of antioxidant superoxide dismutase (Cu/ZnSOD) in presence of P2X4 in HCV infected cells when compared with control NV/HCV. However, expression of both antioxidants was observed unaltered in cells harvested on day 9 of post infection. Results were obtained from 3 individual experiments with replicates sample in each experiment are shown. **Figs 13 and 14.**

## P2X4 in induction of various cytokines

Level of vasoactive cytokine angiotensin II significantly increases in HCV infected cells in presence of P2X4 on day 5 and day 9 when compared with control NV/HCV cells. **Figs 15 and 16.**

Fig 15 shows a significant increase in gene expression of pro-inflammatory cytokine TNF-α and pro-fibrotic cytokine TGF-β in HCV infected cells in presence of P2X4 on day 9 of post infection when compared with HCV infected cells in the absence of P2X4 (Control NV/HCV cells). Results were obtained from 3 individual experiments with replicates sample in each experiment are shown.

**P2X4 in induction of extracellular matrix (ECM) proteins.** A significant increase was observed in levels of ECM markers, elastin and laminin on day 5, 9 of post infection in HCV infected cells in presence of P2X4 than the control NV/HCV cells. Results were obtained from 3 individual experiments with replicates sample in each experiment are shown. **Figs 17 and 18.**

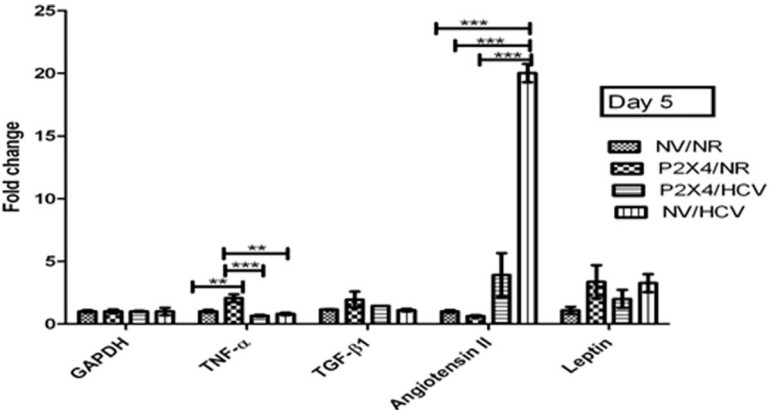

**Fig 15. Expression of various cytokines on day 5 of post infection.** All values are expressed as mean ± SEM, *P≤0.05 vs. control NV/HCV.

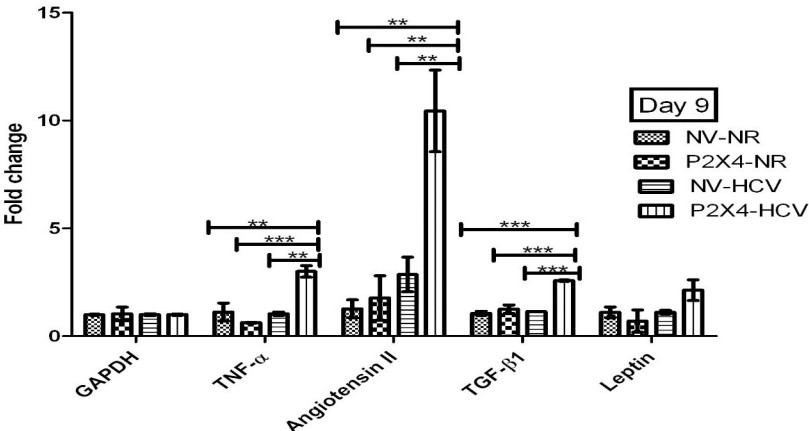

**Fig 16. Expression of various cytokines on day 9 of post infection.** All values are expressed as mean ± SEM, *P≤0.05 vs. control NV/HCV.

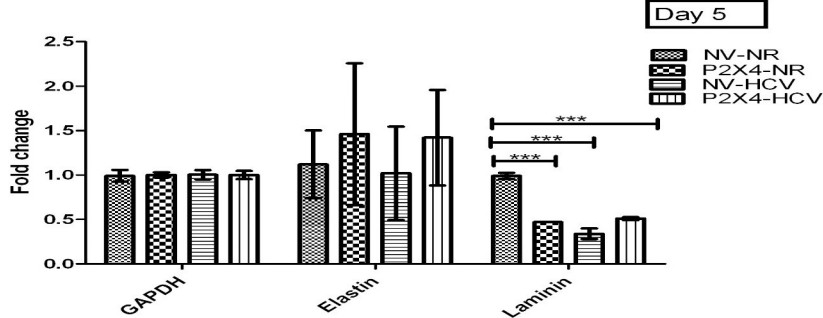

**Fig 17. Expression of ECM markers on day 5 of post infection.** All values are expressed as mean ± SEM, *P≤0.05 vs. control.

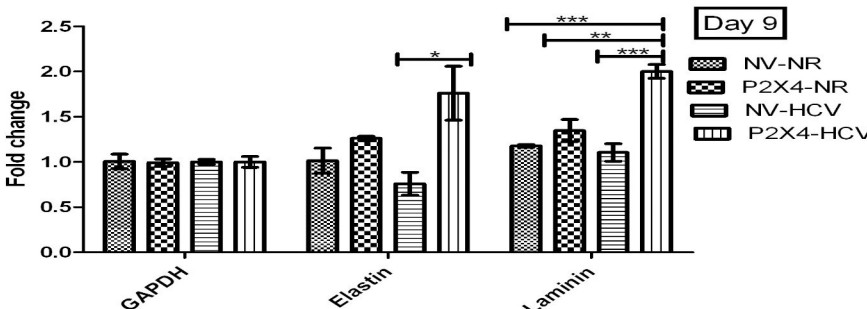

**Fig 18. Expression of various cytokines on day 9 of post infection.** All values are expressed as mean ± SEM, *P≤0.05 vs. control NV/HCV.

## Discussion

Different isoforms of P2X receptors are expressed widely in the body [25–30]. These receptors have known to have their role in inflammation and fibrosis in different organ systems caused

by a number of external stimuli. However, role of P2 receptor in HCV induced inflammation and fibrosis is still under investigation.

There has been a unique pharmacological profile already observed in the brain tissue particularly for P2X4 receptor [31]. A variety of previous reports have supported evidence of presence of P2X4 isoform in liver cells that have important role in regulation of functions such as regulation of hepatic glycogen metabolism *in vivo* mouse models [32], in modulating biliary secretion in rat cholangiocytes. In our previous study, a significant up regulation or activation in gene expression of P2X4 in response to HCV structural proteins E1E2 showed these receptors could be of great importance in HCV pathogenesis [33–35]. So, these observations prompted for the establishment of a cell culture-based system stably expressing the full length P2X4 protein and vector alone (control) so that the role of P2X4 in regulation of genes involved in inducing liver fibrosis could be investigated. Cell lines stably expressing P2X4 proteins and expression vector alone (control) were established.

Buck et al. have successfully established the in vitro replication of HCV RNA after infecting the human HCV sera to human hepatocytes cultures. Furthermore, they detected different levels of structural and nonstructural proteins of virus in infected cultures after 24 hours of infection [36]. In present study, stable cell lines were infected with human HCV RNA (local isolates with genotype 3a) so that role of P2X4 could be studied in the presence of whole HCV genome (HCV replicating RNA). Four different groups P2X4/NR (293T stably expressing P2X4 protein and inoculated with control or normal human sera), NV/NR, (293T stably transfected with vector alone or null vector and inoculated with control or normal human sera), P2X4/HCV (293T stably expressing P2X4 protein and infected with human HCV sera) NV/HCV (293T stably transfected with vector alone or null vector and infected with human HCV sera) were established and successfully maintained till day 5 and day 9 post infection. Cells infected with HCV sera showed replicating HCV RNA till day 9 of infection as shown in the Figs 11 and 12.

P2X4 receptor have shown a significant increase in HCV infected cells (P2X4/HCV) to compare with P2X4 cells in the absence of HCV (P2X4/NV cells). Previous studies have shown that expression of HO-1 increases in response to an acute injury in hepatocytes [37–41].

There was robust up regulation of HO-1 in P2X4 cell line in presence of HCV as compared to all other groups. The present study is in concordance with these reports that HO-1 gene expression was significantly up regulated in HCV infected cells. Whereas, Cu/ZnSOD gene expression was found significantly down regulated in HCV infected P2X4/HCV cells in comparison with NV/HCV group which is HCV infected without P2X4 Figs 13 and 14. The first antioxidant defensive enzymes in the detoxification of oxygen free radicals are the SODs (MnSOD and Cu/ZnSOD). SODs catalyse the dismutation of superoxide radicals into hydrogen peroxide and molecular oxygen [42, 43]. Levent *et al* had examined the serum level of CuZnSOD in chronic HCV patients. They observed that Cu/ZnSOD was significantly lower in erythrocytes of patients with chronic HCV compared with the control [44]. Findings of Elchuri *et al* provide ample evidence that Cu/ZnSOD deficiency in the liver leads to damage to DNA, proteins, and lipids [45]. Long term reduced level of Cu/ZnSOD can be both a potent initiator and promoter in hepatocarcinogenesis. They showed that long term effects of CuZnSOD deficiency in mutant mice lead to widespread oxidative DNA damage which leads to cell injury, cell death, perturbance in cellular homeostasis and eventually to the development of hepatocellular carcinoma in more than 70% *Sod1-/-* mice [45]. Different isoforms of P2X receptors have been identified in different cancer types, in both primary samples of human cancer tissue and human cell lines. The isoforms of P2X receptors mediate proliferation, differentiation and apoptosis [30]. The findings of present study suggest that P2X4 receptors seem to be contributing to HCV induced oxidative stress which is associated with low level of key antioxidant Cu/ZnSOD.

Surprisingly, on the other hand, P2X4 exhibited a positive impact as the level of HO- 1was significantly upregulated in P2X4/HCV infected cells. Liver cells respond to the acute injury with an elevated expression of HO- 1 [40]. It has been reported in literature that induction of HO-1, by chemical inducers or selective overexpression, is cytoprotective both in vitro and in vivo [40]. In this study P2X4 appears to augment the antioxidant response to acute injury in HCV infected cells by upregulating the gene expression of HO-1. There are no previous reports in literature indicating the involvement of P2X receptors in regulation the antioxidant enzymes and the present study identifies the significant role of P2X4 receptor in regulation of key antioxidants in response to HCV replicating RNA in vitro. Clinical results on radiation-induced fibrosis have shown that CuZnSOD significantly reduces the expression of the TGF-β, both at the mRNA and protein leves [46]. Consistent with this, significant upregulated CuZn-SOD expression in NV/HCV cells showed a significant reduced TGF-β expression in this group as depicted in Figs 13 and 14.

Transforming growth factor-beta (TGF-β) is reported to be the main profibrotic factor in liver fibrosis [47–49]. It has been reported in literature that TGF-β was upregulated in HCV infection and induces matrix accumulation by activating hepatic stellate cells [49]. Previous studies provide evidences that among the different cellular pathways involved in fibrosis, the transforming growth factor-β1 (TGF-β1) signaling plays a critical role. TGF- β is a key regula-tor for extracellular matrix proteins (ECM) metabolism and functions as an autocrine and a paracrine mediator. The impact of TGF- β on liver fibrosis has been well documented in TGF-β knockout mouse model [50], in attenuating the development of liver fibrosis by using soluble type II TGF-β receptor [51].

Activation of P2X7 receptors (isoform of P2X receptors) increased TGF-β mRNA expres-sion in type-2 rat brain astrocytes [27]. Goncalves et al studied the role of P2X receptors in inflammatory and fibrogenic response in the kidneys of unilateral ureteral obstruction (UUO) [52]. They used P2X7 knockout mice (-/-). They found significant reduced TGF-β expression in UUO P2X7 (-/-) mice comared to UUO WT mice. In this study, TGF-β gene expression was also analysed in all 4 groups. It is noteworthy that profoundly increased gene expression of P2X4 in HCV infected cells (P2X4/HCV) is accompanied with significantly elevated expres-sion of TGF-β in P2X4/HCV cells compared to all other groups as shown in Figs 15 and 16. Thus, it demonstrates the involvement of P2X4 in the production of this profibrotic cytokine in P2X4/HCV infected cells.

Previous studies provide evidences that angiotensin II (Ang II) plays a pivotal role in the progression of chronic liver diseases, i.e., liver fibrosis and hepatocellular carcinoma. Previous reports demonstrate that Ang II is found to be frequently activated in HCV patients [53]. The present study showed an increased Ang II expression in HCV infected cells, NV/HCV and P2X4/HCV (Figs 13 and 14). Paizis et al reported that the blockage of Ang II inhibits TGF-β expression in experimental liver fibrosis [54]. Findings of Paizis et al were confirmed by Batal-ler et al, showed that Ang II increased TGF-β1 mRNA expression in cultured rat hepatic stel-late cells [55]. Livers from bile duct-ligated rats infused with Ang II showed increased TGF-β content [56]. The current study is in agreement with these previous observations and reveals a significantly elevated TGF-β mRNA expression in P2X4/HCV group on day 9 post infection along with profound increased gene expression of Ang II as shown in Further, the present study demonstrates significant role of P2X4 in regulation of Ang II. A dramatic increase was observed in gene expression of Ang II in NV/ HCV cells comared to P2X4/HCV cells on day 5 post infection (shown in Figs 15 and 16). However at day 9 post infection a remarkable signifi-cant increase was observed in gene expression of Ang II in P2X4/ HCV cells in comparison with NV/HCV cells. This increase was accompanied with a significant elevated TGF-β gene expression in P2X4/ HCV group in comparison to NV/HCV group. It is noteworthy here that

the significant increase in TGF-β mRNA expression along with significant raised level of Ang II in HCV infected cells is associated with the presence of P2X4. Thus the present data suggests the possible role of P2X4 (purinergic signaling) in upregulation of this important vasoactive, profibrotic cytokine, Ang II in HCV infected cells. There is no data already available in literature concerning the involvement of P2X receptors in the gene expression of Ang II. Thus the current study identifies the additional role of P2X receptors to fibrogenic responses.

TNF-α is a potent pro-inflammatory cytokine, having multiple biological activities [57]. The level of TNF-α was found to be elevated in patients with cirrhosis or acute/chronic hepatitis when compared with healthy patients in different studies [58, 59]. Increased level of TNF-α is associated with hepatic inflammation, necrosis and hepatic failure [60, 61]. In a study by Batellar *et al* Ang II infusion in rats undergoing biliary fibrosis, increased serum levels of Ang II and bile duct ligation-induced liver injury. It augmented the hepatic concentration of inflammatory proteins (TNF-α, IL-1β) [56]. In this study a significant increase in gene expression of TNF-α and Ang II in P2X4/ HCV cells was observed in comparison to NV/HCV cells on day 9 post infection. Liver inflammation is the hallmark of early-stage liver fibrosis and eventually results in activation of hepatic stellate cell (HSC) and extracellular matrix (ECM) deposition [62].

In addition the present study demonstrates that the significant increase in TNF-α gene expression along with significantly elevated level of Ang II in P2X4/HCV infected cells is coupled with elevated expression of P2X4 (Figs 15 and 16). Surprisingly, a significantly increased TNF-α gene expression was observed in P2X4/NR cells on day 5 post infection in comparison with other groups, NV/NR cells, P2X4/HCV cells, NV/HCV cells as shown in Figs 15 and 16.

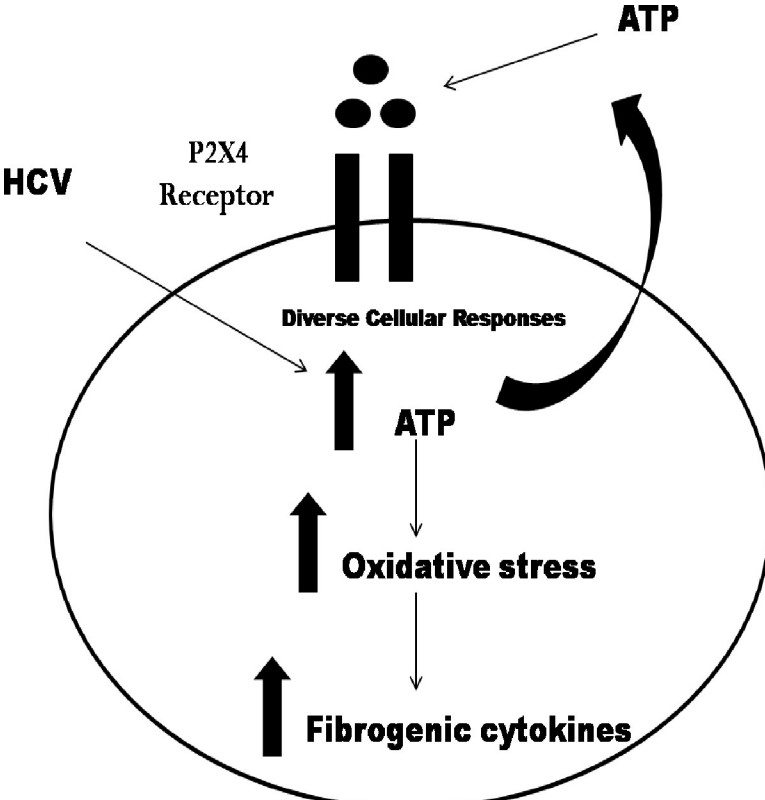

**Fig 19. Schematic overview of the purinergic signaling in the presence of HCV.** HCV interrupts purinergic signaling Increased ATP in the extracellular mileu increases oxidative stress and fibrogenic cytokines and ECM cytokines leading towards fibrosis.

This is consistent with previous studies which illustrate the involvement of P2X receptor activation (P2X7) in inflammation [63] and elimination of intracellular pathogens [64]. Several studies have addressed the role of P2X receptors activation on the mechanisms of immunological and inflammatory response of Leukocytes and monocytes/macrophages [65]. No previous reports are found in literature demonstrating the involvement of P2X4 isoform of P2X receptors in release of inflammatory cytokine. The current study identifies the role of P2X4 isoform of P2X receptors (P2X4) in upregulation of pro inflammatory cytokine TNF-α. Leptin, an adipocytokine is documented to regulate liver fibrogenesis. High serum leptin concentrations are present in chronic HCV patients than controls [66] and is associated with severity of fibrosis [67]. In contrast, another study revealed that there is no correlation between leptin level and fibrosis in chronic HCV patients [67]. The present study observed an ultered gene expression of leptin in all groups, NV/NR cells, P2X4/NR cells, P2X4/HCV cells and NV/HCV cells on day 5 and day 9 post infection as shown in Figs 15 and 16.

The current study indicated a significant increase in gene expression of elastin and laminin (extracellular matrix markers) in P2X4/HCV cells compared to NV/HCV cells on day 9 HCV infection as shown in Figs 17 and 18. The significant increase in mRNA expression of elastin and laminin in the presence of P2X4 in response to HCV infection suggests the involvement of P2X4 in this regard. The role of P2X receptors in matrix deposition in response to HCV induced inflammation has not been reported in literature previously.

Thus, the present study is the first one to identify the involvement of P2X receptors (P2X4) in upregulation of gene expression of extracellular matrix markers. The results of this study suggested the involvement of P2X4 receptors in HCV induced oxidative stress, inflammation, and fibrogenic responses and matrix deposition. This study provides important clues to the mechanisms involved in the progression of chronic liver disease with the potential of hepatocellular carcinoma (Fig 19).

## Supporting information

**S1 File. Sequence homology of full length P2X4 sequence.** (1.7-kb) (Rattus norvegicus), kindly provided by Dr. Ishtiaq Qadri with reported sequence (Sequence chromatogram not shown).
(PDF)

**S2 File. Representative chromatograms of sequence of transfected P2X4 receptor amplified from stable cell line (293T/P2X4).**
(PDF)

**S3 File. Sequence homology of transfected P2X4 (stable cell line 293T/P2X4) with sequence kindly provided by Dr.** Ishtiaq Qadri along with full length gene P2X4 (1.7-kb).
(PDF)

**S4 File. 293T/P2X4 cell line experimental and control in presence of G418 (a, b.** 100X), 293T/NV cell line experimental and control in presence of G418(c, d. 100X). 293T/P2X4 cell line and 293T/NV cell line demonstrated better resistance to G418 in comparison to both control cell lines resulting in increased proliferation of growth.
(PDF)

**S5 File.**
(PDF)

**S1 Data.**
(ZIP)

## Author Contributions

**Data curation:** Sobia Manzoor.

**Formal analysis:** Sobia Manzoor.

**Methodology:** Sobia Manzoor.

**Project administration:** Muhammad Idrees.

**Software:** Sobia Manzoor.

**Validation:** Sobia Manzoor.

**Writing – review & editing:** Madiha Khalid.

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
