## [Decision Letter · Decision Letter 0]

29 Nov 2021

PONE-D-21-33744P2x4 Receptors in Presence of Successfully Replicating HCV Mediate Induction of Antioxidants, Cytokines and ECM Transcripts; An Insight into role of P2X4 in fibrosisPLOS ONE

Dear Dr. Sobia Manzoor,

Thank you for submitting your manuscript to PLOS ONE. After careful consideration, we feel that it has merit but does not fully meet PLOS ONE’s publication criteria as it currently stands. Therefore, we invite you to submit a revised version of the manuscript that addresses the points raised during the review process.

We look forward to receiving your revised manuscript.

Kind regards,

Usman Ali Ashfaq, PhD

Academic Editor

PLOS ONE

Journal Requirements:

"Higher Education commission Pakistan for financial support."

"nO.The funders had no role in study design, data collection and analysis, decision to publish, or preparation of the manuscript."

7. Please amend either the abstract on the online submission form (via Edit Submission) or the abstract in the manuscript so that they are identical.

8. Please upload a copy of Figure 34 and 35 to which you refer in your text on page 15. If the figure is no longer to be included as part of the submission please remove all reference to it within the text.

Additional Editor Comments:

The article has been accepted for publication after incorporation of minor comments raised by reviewer

Reviewers' comments:

Reviewer's Responses to Questions

**Comments to the Author**

1. Is the manuscript technically sound, and do the data support the conclusions?

Reviewer #1: Yes

Reviewer #2: Yes

2. Has the statistical analysis been performed appropriately and rigorously? 

Reviewer #1: Yes

Reviewer #2: Yes

3. Have the authors made all data underlying the findings in their manuscript fully available?

Reviewer #1: Yes

Reviewer #2: Yes

4. Is the manuscript presented in an intelligible fashion and written in standard English?

Reviewer #1: Yes

Reviewer #2: Yes

5. Review Comments to the Author

Reviewer #1: Study entitled “P2x4 Receptors in Presence of Successfully Replicating HCV Mediate Induction of Antioxidants, Cytokines and ECM Transcripts; An Insight into role of P2X4 in fibrosis” shows a notable effort by Manzoor et al., towards understanding the molecular mechanmisms underlying the HCV inection and pathogenesis and pathophysiology of P2X receptors. It is an exciting and new insight in the field of HCV research. It demonstrates the involvement of the P2X4 receptors in HCV induced pathogenesis. This study opens a new insight into HCV associated pathologies and would be a good and useful addition in existing literature. This study could be published after incorporating the suggested changes.

Major Points:

1. As there are different purinergic receptors, what was the reason to select only P2X4? And why different P2X receptors were not targeted for the determination of their role in liver fibrosis? Please address and also incorporate in manuscript.

2. Why did you use HEK for the transfection of the P2X4 clone? Why not be tested in any other cell line?

3. As I went through, I didn’t find any oxidant levels measured in the findings. Would you explain the reason for that?

Minor Points

The text contains several orthographic and grammatical errors, including missing spaces (often between numbers and the corresponding unit of measure) and incomplete sentences.

1. Page 23, Kindly explain the abbreviations (cy3 and fam) in the formula.

2. Page 17, Rephrase the first sentence.

3. Page 17, last paragraph It seems that day may not agree in number with other words in this phrase.

4. Page 19, replace complete medium

5. Page 21, Rephrase the sentence in the first paragraph

6. Page 22 explain the term BCIP/ NBT solution

7. Explain heading 4.5, 4.6 and 4.7 in the results section.

8. Correct spelling mistakes in the overall manuscript.

Reviewer #2: The study is an effort towards finding the role of P2X4 receptors in the presence of HCV. It explains how HCV plays a role in the regulation of key antioxidant enzymes (HO-1, Cu/ZnSOD) in the induction of proinflammatory cytokine (TNF-α), major profibrotic cytokine (TGF-β) and vasoactive cytokine (angiotensin II) through these receptors. Study also reveal the role of the P2X4 receptor in increasing the expression of extracellular matrix proteins in the presence of HCV. The current study is a good attempt to decipher the involvement of secondary signaling in HCV pathogenesis. However, authors are required to address the following points before formal acceptance of the manuscript.

Title:

Reconsider the tile as it isn’t suitable. There is a lot of mistakes in it. It can be changed to make the manuscript catchy.

1. Abstract

Well written.

2. Introduction

a- Reconsider the typographic and grammatically changes in the manuscript.

b- In the introduction last paragraph authors have discussed the two vectors and HCV sera. Can they clear it which two vectors as they have only discussed only P2X4 vector?

c- Why they need to do over expression studies by inducing the vector of gene externally when they want to see the effect of HCV sera on the expression of P2X4 protein?

d- Is there any reference available of HCV replication in H293T cells?

3. Material and methods

a- Brief the transfection method and selection of stable cell lines. Readjust sample collection and sera inoculation.

b- Can authors specify any specific reason to clone the gene from one mammalian expression vector (pCR3.1) to another expression vector (pcDNA3.1)?

c- In heading 3.7., Line ‘Following primers were used for PCR amplifications from cDNA.” there is no primer list given provide them.

d- In heading 3.8, line, ‘Amplified product of expected size was obtained” what was the expected size? mention it.

e- In heading 3.10, what was protein isolation procedure? T what passage or days post transfection protein was isolated? Mention it.

f- What was the range of viral titer used for inoculation experiments?

g- Cite the tables and figures in the main text where first used.

h- Please mentioned which HCV genotype serum was used?

4. Results

a. Why was there no expression checked for fibrotic markers MMP-9?

b. Why was cell volume regulation neglected in the presence of HCV?

c. Does P2X7 receptor was not assessed for its role in liver fibrosis?

d. Rewrite the figure legends and improve them as they lack explanation.

e. Why was angiotensin selected as a cytokine?

f. Elaborate the results. If authors are mentioning it is significant, they should write either fold change or level of significance.

g. Quality of figures need to be improved. The text written in Fig 1b is not readable.

h. Figure 1c, Figure 2 and 3 the labeling of the gel pics is not visible. Authors need to improve it.

i. The gel pictures where expression is measured through Real Time PCR should be provided as supplementary material not in main article.

j. Authors did inoculation experiment in stably expression P2X4 gene. Are authors performed any expression studies in non-stable cell line and effect of P2X4 gene in inoculation experiment.

k. In western blot experiment authors are encouraged to measure the expression through any software like Image J etc.

l. Remove typo and grammatical mistakes through out the document.

Conclusion: If authors give a pictorial form of their conclusion section it would be clearer.

6. PLOS authors have the option to publish the peer review history of their article (what does this mean?). If published, this will include your full peer review and any attached files.

Reviewer #1: **Yes: **Dr Saba Khaliq

Reviewer #2: No

---

## [Author Response · Author response to Decision Letter 0]

14 Feb 2022

Journal Requirements

1. Please ensure that your manuscript meets PLOS ONE's style requirements, including those for file naming?

Authors’ response.The manuscript and file names have been updated according to the style requirements of PLOS ONE.

Authors’ response .We are thankful your suggestion. All the references has been cross verified. There are no such references included that are retracted. 

Authors’ response. Subject study was financed by student research funds provided to student by University as well as Higher Education Commission (HEC) of Pakistan.

Thank you for stating the following in the Acknowledgments Section of your manuscript: "Higher Education commission Pakistan for financial support."We note that you have provided funding information that is not currently declared in your Funding Statement. However, funding information should not appear in the Acknowledgments section or other areas of your manuscript. We will only publish funding information present in the Funding Statement section of the online submission form. 

Please remove any funding-related text from the manuscript and let us know how you would like to update your Funding Statement. Currently, your Funding Statement reads as follows: "no”. The funders had no role in study design, data collection and analysis, decision to publish, or preparation of the manuscript."Please include your amended statements within your cover letter; we will change the online submission form on your behalf.

Authors’ response. All the funding was provided to student byUniversity as well as Higher Education Commission (HEC) of Pakistan for smooth and successful completion of current study.

4. In your Data Availability statement, you have not specified where the minimal data set underlying the results described in your manuscript can be found. PLOS defines a study's minimal data set as the underlying data used to reach the conclusions drawn in the manuscript and any additional data required to replicate the reported study findings in their entirety. All PLOS journals require that the minimal data set be made fully available. For more information about our data policy, please see http://journals.plos.org/plosone/s/data-availability. Upon re-submitting your revised manuscript, please upload your study’s minimal underlying data set as either Supporting Information files or to a stable, public repository and include the relevant URLs, DOIs, or accession numbers within your revised cover letter. For a list of acceptable repositories, please see http://journals.plos.org/plosone/s/data-availability#loc-recommended-repositories. Any potentially identifying patient information must be fully anonymized. Important: If there are ethical or legal restrictions to sharing your data publicly, please explain these restrictions in detail. Please see our guidelines for more information on what we consider unacceptable restrictions to publicly sharing data: http://journals.plos.org/plosone/s/data-availability#loc-unacceptable-data-access-restrictions. Note that it is not acceptable for the authors to be the sole named individuals responsible for ensuring data access.

Authors’ response. All data that is included in this manuscript has been taken from the doctoral dissertation of first and corresponding author SobiaManzoor. Furthermore, corresponding author and her parent institution where the major part of subject study was conducted (Centre of Excellence in Molecular Biology, University of The Punjab, Lahore, Pakistan) have record of all data of current study on Laboratory data recording books as well as in the doctoral dissertation of first and corresponding author, SobiaManzoor.

5. PLOS ONE now requires that authors provide the original uncropped and unadjusted images underlying all blot or gel results reported in a submission’s figures or Supporting Information files. This policy and the journal’s other requirements for blot/gel reporting and figure preparation are described in detail at https://journals.plos.org/plosone/s/figures#loc-blot-and-gel-reporting-requirements and https://journals.plos.org/plosone/s/figures#loc-preparing-figures-from-image-files. When you submit your revised manuscript, please ensure that your figures adhere fully to these guidelines and provide the original underlying images for all blot or gel data reported in your submission. See the following link for instructions on providing the original image data: https://journals.plos.org/plosone/s/figures#loc-original-images-for-blots-and-gels. In your cover letter, please note whether your blot/gel image data are in Supporting Information or posted at a public data repository, provide the repository URL if relevant, and provide specific details as to which raw blot/gel images, if any, are not available. Email us at plosone@plos.org if you have any questions.

6. Authors’ response. Any part of the data that is presented in this manuscript has never been published in any public repository.

7. Please amend either the abstract on the online submission form (via Edit Submission) or the abstract in the manuscript so that they are identical

Authors’ response .We are thankful for your suggestion. We will incorporate the changes in revised manuscript.

8. Please upload a copy of Figure 34 and 35 to which you refer in your text on page 15. If the figure is no longer to be included as part of the submission please remove all 

Authors’ response.We appreciate your suggestion and the said changes have been incorporated.

Response to Reviewers’ Comments

Reviewer #1: Study entitled “P2x4 Receptors in Presence of Successfully Replicating HCV Mediate Induction of Antioxidants, Cytokines and ECM Transcripts; An Insight into role of P2X4 in fibrosis” shows a notable effort by Manzoor et al., towards understanding the molecular mechanmisms underlying the HCV inection and pathogenesis and pathophysiology of P2X receptors. It is an exciting and new insight in the field of HCV research. It demonstrates the involvement of the P2X4 receptors in HCV induced pathogenesis. This study opens a new insight into HCV associated pathologies and would be a good and useful addition in existing literature. This study could be published after incorporating the suggested changes.

Major Points:

1. As there are different purinergic receptors, what was the reason to select only P2X4? And why different P2X receptors were not targeted for the determination of their role in liver fibrosis? Please address and also incorporate in manuscript.

Author’s Response: As the study published by SobiaManzooret al, 2011 it was alreadyestablished in our lab that the gene expression of identified isoforms of P2X receptors in presence of HCV structural proteins E1E2, Huh-7/E1E2 cell line (stably expressing HCV structural proteins E1E2). These findings provide molecular evidence that P2X receptors are also present in human liver cells (Huh-7 cell line), and indicated that P2X receptors respond towards HCV structural proteins E1E2 of genotype 3a, P2X4 is one of the most responsive isoforms. 

2. Why did you usedthe HEK for the transfection of the P2X4 clone? Why not be tested in any other cell line?

Author’s Response: All other cell lines fully express P2X4 receptors whereas HEK-293 cells have least expression of purinergic receptors so there was need of cell line that do not already express the receptors to avoid the false positive results.

3. As I went through, I didn’t find any oxidant levels measured in the findings. Would you explain the reason for that?

Author’s Response: As the antioxidant levels were measured and it’s already reported in various findings that oxidant levels are increased during HCV infection. Muhammad Y.Sheikh ; 2008 et al.

Minor Points

The text contains several orthographic and grammatical errors, including missing spaces (often between numbers and the corresponding unit of measure) and incomplete sentences.

1. Page 23, Kindly explain the abbreviations (cy3 and fam) in the formula.

Author’s Response:Fluorescence is observed in Real Time on the Cy3 channel for HCV RNA and FAM channel for Internal Control.

2. Page 17, Rephrase the first sentence.

Author’s Response:We are thankful for your suggestion. The said changes have been incorporated.

3. Page 17, last paragraph It seems that day may not agree in number with other words in this phrase.

Author’s Response: We appreciate your response for the suggested changes. All the changes have been incorporated and highlighted in the manuscript.

4. Page 19, replace complete medium

Author’s Response: The said changes have been incorporated

5. Page 21, Rephrase the sentence in the first paragraph

Author’s Response: Sentence has been rephrased. 

6. Page 22 explain the term BCIP/ NBT solution

Author’s Response:5-bromo-4-chloro-3-indolyl-phosphate), 4-nitro blue tetrazolium chloride

7. Explain heading 4.5, 4.6 and 4.7 in the results section.

Author’s Response: This was an appropriate suggestion that have been incorporated. 

8. Correct spelling mistakes in the overall manuscript.

Author’s Response: All the spelling mistakes have been changed. 

Reviewer #2: The study is an effort towards finding the role of P2X4 receptors in the presence of HCV. It explains how HCV plays a role in the regulation of key antioxidant enzymes (HO-1, Cu/ZnSOD) in the induction of proinflammatory cytokine (TNF-α), major profibrotic cytokine (TGF-β) and vasoactive cytokine (angiotensin II) through these receptors. Study also reveal the role of the P2X4 receptor in increasing the expression of extracellular matrix proteins in the presence of HCV. The current study is a good attempt to decipher the involvement of secondary signaling in HCV pathogenesis. However, authors are required to address the following points before formal acceptance of the manuscript.

Title:

Reconsider the tile as it isn’t suitable. There is a lot of mistakes in it. It can be changed to make the manuscript catchy.

1. Abstract

Well written.

2. Introduction

a- Reconsider the typographic and grammatically changes in the manuscript.

Author’s Response: All the typographic changes have been addressed.

b- In the introduction last paragraph authors have discussed the two vectors and HCV sera. Can they clear it which two vectors as they have only discussed only P2X4 vector?

Author’s Response:The line has been rephrased.it’s not about two vectors the word both vectors refer to the vector and sera.

c- Why they need to do over expression studies by inducing the vector of gene externally when they want to see the effect of HCV sera on the expression of P2X4 protein?

Author’s Response: As HEK cell line do not express purinergic protein, the basic idea of the study was to see the effect on P2X4 receptors in the presence of HCV.

d- Is there any reference available of HCV replication in H293T cells?

Author’s Response: There is no reference available for the replication of HCV in HEK 293 cells. Present study reports the in vitro replication of HCV in HEK 293 cells first time in literature. Established evidence suggested the replication of HCV in peripheral blood mononuclear cells (PBMCs) besides hepatic cells in patients chronically infected with HCV. 

3. Material and methods

a- Brief the transfection method and selection of stable cell lines. Readjust sample collection and sera inoculation.

Author’s Response: The authors acknowledge the fact that methodology should be easily understood by the reader hence necessary changes have been made where required.

b- Can authors specify any specific reason to clone the gene from one mammalian expression vector (pCR3.1) to another expression vector (pcDNA3.1)?

Author’s Response:pcDNA3.1 is constructed to study the expression of gene of interest in mammalian expression system. That’s why gene of interest, P2X4 was sub-cloned from vector(pCR3.1) toanother expression vector (pcDNA3.1) .

c- In heading 3.7., Line ‘Following primers were used for PCR amplifications from cDNA.” there is no primer list given provide them.

Author’s Response: This suggestion is appropriate, as list was referred in the supplementary files. The line has moved from the section.

d- In heading 3.8, line, ‘Amplified product of expected size was obtained” what was the expected size? Mention it.

Author’s Response:We are grateful to you for pointing out the error. The line has been moved as the size has been discussed in the result section. 

e- In heading 3.10, what was protein isolation procedure? What passage or days post transfection protein was isolated? Mention it.

Author’s Response:Since it was stable expression not a transient expression. Protein was isolated after 96 hours of post transfection. 

f- What was the range of viral titer used for inoculation experiments?

Author’s Response:

g- Cite the tables and figures in the main text where first used.

Author’s Response:we are thankful for your suggestion. All the figures have been incorporated at right place.

h- Please mentioned which HCV genotype serum was used?

Author’s Response: The genome used was HCV genotype 3a.

4. Results

a. Why was there no expression checked for fibrotic markers MMP-9?

Author’s Response: Your suggestion is appropriate. As we measured other fibrotic genes but it can be incorporated in our future studies.

b. Why was cell volume regulation neglected in the presence of HCV?

Author’s Response: We appreciate your thoughtful suggestion, but it was out of scope of our study’s design. Our study design was focused to investigate the role of P2X4 in HCV associated liver fibrosis only. 

c. Does P2X7 receptor was not assessed for its role in liver fibrosis?

Author’s Response: P2X4 receptor role was more prominent and significant while P2X7 studies have been done for their presence in PBMCs in the blood of chronic HCV patients. (World J Gastroenterol. 2003 Feb 15; 9(2): 291–294. Published online 2003 Feb 15. doi: 10.3748/wjg.v9.i2.291 PMCID: PMC4611331).

d. Rewrite the figure legends and improve them as they lack explanation.

Author’s Response: The suggestions incorporated in the respective section.

e. Why was angiotensin selected as a cytokine?

Author’s Response: The current study included fibrogenic cytokines TGF, angiotensin and leptin. 

f. Elaborate the results. If authors are mentioning it is significant, they should write either fold change or level of significance.

Author’s Response: The said changes have been incorporated.

g. Quality of figures need to be improved. The text written in Fig 1b is not readable.

Author’s Response: we appreciate your suggestion and have been incorporated.

h. Figure 1c, Figure 2 and 3 the labeling of the gel pics is not visible. Authors need to improve it.

Author’s Response: We appreciate your suggestion. The said changes have been incorporated. 

i. The gel pictures where expression is measured through Real Time PCR should be provided as supplementary material not in main article.

Author’s Response: All the said suggestions have been incorporated. 

j. Authors did inoculation experiment in stably expression P2X4 gene. Are authors performed any expression studies in non-stable cell line and effect of P2X4 gene in inoculation experiment.

Author’s Response:No, current study was designed to evaluate whether P2X4 plays any role in HCV associated liver fibrosis. For this purpose, a cell line with stable expression of P2X4 was the requirement. 

k. In western blot experiment authors are encouraged to measure the expression through any software like Image J etc.

Author’s Response:We appreciate your suggestion. Since the hypothesis of the study was to investigate the probable role of P2X4 in HCV associated pathogenesis (Fibrosis) in the presence of P2X4 only. That’s why we did not estimate the expression of P2X4.

l. Remove typo and grammatical mistakes throughout the document.

Author’s Response: All the said changes have been incorporated.

Conclusion: If authors give a pictorial form of their conclusion section it would be clearer.

---

## [Editor Report · Decision Letter 1]

3 Mar 2022

P2X4 receptors mediate induction of antioxidants, fibrogenic cytokines and ECM transcripts; in presence of replicating HCV in in vitro setting:  an insight into role of P2X4 in fibrosis

PONE-D-21-33744R1

Dear Dr. Sobia Manzoor

We’re pleased to inform you that your manuscript has been judged scientifically suitable for publication and will be formally accepted for publication once it meets all outstanding technical requirements.

Kind regards,

Usman Ali Ashfaq, PhD

Academic Editor

PLOS ONE

Additional Editor Comments (optional):

Acceptable for publication in plos one in current form
---

## [Editor Report · Acceptance letter]

5 Apr 2022

PONE-D-21-33744R1 

P2X4 receptors mediate induction of antioxidants, fibrogenic cytokines and ECM transcripts; in presence of replicating HCV in *in vitro* setting:  an insight into role of P2X4 in fibrosis  

Dear Dr. Manzoor:

I'm pleased to inform you that your manuscript has been deemed suitable for publication in PLOS ONE. Congratulations! Your manuscript is now with our production department. 

Kind regards, 

on behalf of

Dr. Usman Ali Ashfaq 

Academic Editor

PLOS ONE